# QUALITY-DIVERSITY TRANSFER LEARNING (QDTL)

## ABSTRACT

Deep learning has had much success on challenging problems with large datasets. However, it struggles in cases with limited training data. Transfer learning represents a class of approaches for transferring knowledge from large source datasets to smaller target datasets. But many transfer learning approaches have constraints in terms of dataset size and similarity of output features. In this paper, we introduce Quality-Diversity Transfer Learning (QDTL), a novel transfer learning approach based on neuroevolution for dealing with very small dataset problems with distinct output features. We demonstrate the success of QDTL on two medical prediction problems, outperforming standard transfer learning baselines.

## 1 INTRODUCTION

Deep learning is currently used to address a wide variety of challenging problems. A large dataset is one of the key prerequisites for modern deep learning techniques to be effective (Sarker, 2021). As such, data limitations can serve as a major bottleneck to the application of deep neural networks (DNNs). Moreover, this problem may be more prevalent in the future, due to data pollution by large language models (Villalobos et al., 2022). Regardless, currently a number of domains are missing out on benefitting from DNNs because they lack sufficient data. Medical is one such domain where data is limited due to costs and privacy concerns.

Currently, low-data machine learning models like linear and logistic regressions are commonly used in the medical domain despite their low accuracy (Shamshirband et al., 2021). More advanced techniques, like DNNs can be applied to such domains via transfer learning; an approach used to reuse pre-trained model to solve problems with less data (Bica & van der Schaar, 2022; Yoon et al., 2022). However, there are certain challenges associated with the effective use of transfer learning approaches. For instance, given a small dataset, they are prone to overfitting, leading to poor generalization. Further, the differences in data collection and variations in output feature space can negatively impact the effectiveness of transfer learning methods (Weiss et al., 2016).

Differences in data collection and variations in the output feature space between source and target domains can hinder the effectiveness of transfer learning methods (Weiss et al., 2016). Additionally, transfer learning models are prone to overfitting, especially when the target dataset is small, leading to poor generalization. Other approaches to learn for a limited target dataset, such as Few-shot or zero-shot approaches are not applicable to our targeted tasks (Section 5) the medical domain due to limited dataset sizes and lack of shared classes. To overcome these limitations and enable the application of DNNs to more medical problems, we propose a novel transfer learning approach called Quality-Diversity Transfer Learning (QDTL).

QDTL combines neuroevolution, architecture search, and conceptual expansion. We utilize quality-diversity (QD) optimization, which directly modifies the weights (parameters) of DNN, enabling higher performance for low-training data tasks. QD optimization considers both the quality and diversity of solutions, potentially mitigating overfitting. While architecture search allows us to automatically discover the optimal neural network architecture for the task at hand. Further, we incorporate the conceptual expansion representation Guzdial & Riedl (2019) to describe the target model as a combination of weights from a source model. This enables us to treat source model knowledge as a combinational creativity task, expanding the capabilities of our approach.

Contributions of this work are highlighted below:

1. A novel transfer learning approach to solve low-data problems via quality-diversity-based neuroevolution: Quality-Diversity Transfer Learning (QDTL)

2. The application of QDTL to the task of predicting pre-term birth, with the ability to predict births within 7 days; significantly improving over the current state of the art.

3. The application of QDTL to predict the survival days of patients undergoing organ transplant; surpassing the finetuning approach traditionally applied to transfer learning medical prediction problems.

## 2 PROBLEM FORMULATION

In this paper, we focus on low-data medical prediction tasks. Formally, the source domain is defined as $S$ with $N_s$ instances, each with $D_s$-dimensional features and $S = \{(x_{si}, y_{si})\}$, where $x_{si} \in \mathbb{R}^{D_s}$, $y_{si}$ represents time until a medical event. The target domain is defined as $T$ with $N_t$ instances, each with $D_t$-dimensional features such that $T = \{(x_{ti}, y_{ti})\}$, where $x_{ti} \in \mathbb{R}^{D_t}$, $y_{ti}$ represents time until a medical event. Our objective is to adapt knowledge from the source domain to the target domain, where $N_s < 140$, $N_t < 200$ and where $D_s$ and $D_t$ can be different with no shared output features. This precludes the application of many zero-shot and few-shot approaches. Our objective is to minimize prediction error on $T$ given the data scarcity and feature dissimilarity.

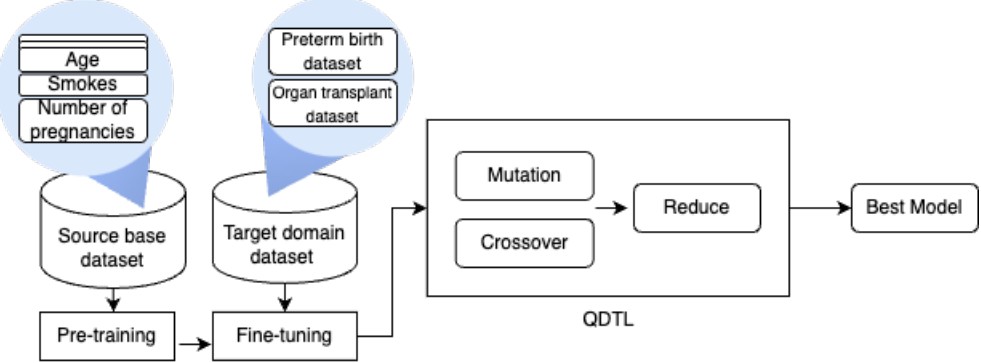

Figure 1: QDTL Process

## 3 RELATED WORK

This section describes related work on evolutionary search, architecture search, transfer learning, and the applications of transfer learning in medical domain.

### 3.1 EVOLUTIONARY SEARCH & ARCHITECTURE SEARCH

Evolutionary algorithms are optimization algorithms inspired by biological evolution. Among them, Quality-Diversity (QD) algorithms aim to provide both optimal and diverse solutions to a given problem. QD algorithms are widely used in fields such as robotics and video games to achieve effective solutions (Mouret & Maguire, 2020; Keller et al., 2020; Gravina et al., 2019). QD algorithms typically work by maintaining a population of candidate solutions, and iteratively improving them through a process of selection, variation, and evaluation.

One of the key characteristics of QD algorithms is their focus on generating diverse solutions, which sets them apart from other optimization algorithms. Despite the success of QD algorithms in various domains, there has been limited work on applying them to transfer learning problems. Salehi et al. (2022) introduced few-shot learning with quality-diversity optimization for learning from unseen tasks. They proposed a model that learns a population of prior policies to initialize a quality-diversity optimization problem for limited data training. Our work shares a similar intuition but focuses on low-data supervised transfer learning instead of reinforcement learning tasks.

Architecture search is a method used in machine learning to automatically discover the optimal neural network architecture for a given task (Floreano et al., 2008). Singamsetti et al. (2021) combined architecture search with transfer learning in the image classification domain. Their work showed success in finding optimal structure using evolutionary-based search. In our work, we have combined QD and evolutionary-based search. Our work is different from the aforementioned work in terms of the optimization approach; focus on regression tasks, and a different setting of the architecture search problem. Our Genetic Algorithm (GA) ablation study 6.1.2 can be seen as an approximation of their approach.

## 3.2 Transfer Learning

Transfer Learning approaches can be categorized into four types: instances-based, mapping-based, network-based, and adversarial-based (Tan et al., 2018). Most of the prior works in transfer learning make use of finetuning approach (Guo et al., 2019). For example, Hu et al. (2021) proposed a finetuning approach that maintained a high-quality model but not requiring full finetuning and only training some dense layers in a neural network indirectly. In our approach, we use a finetuned model as our initial model and further perform optimization in order to overcome the overfitting problem. Moreover, we are using architecture search so this is not useful for our case.

Zhang et al. (2020) combined neuroevolution and transfer learning approach. They proposed a new approach for adapting and transferring deep neural networks (DNNs) using a hybrid evolution strategy (HES). The HES combines the strengths of both genetic algorithms and particle swarm optimization to efficiently explore and optimize the search space of DNN architectures. Their approach is specific to image classification tasks and would be difficult to generalize it for other tasks.

In the most related example of prior work, Mahajan & Guzdial (2022) used a tree-based transfer learning approach to learn the behaviour of an individual on a target task using the data of that same individual available on a secondary task in the financial domain. Similar to their approach, we are using a finetuned model as an input. They are using a Monte Carlo Tree Search(MCTS) optimization approach which would not be helpful in our case as we don't have sequential data and neither are our source and target tasks related.

## 3.3 Transfer Learning Applications in the Medical Domain

Transfer learning is an approach that adapts existing knowledge from one domain to another. Transfer learning is beneficial in the medical domain by aiding in resolving the issue of data scarcity, and a common approach is to finetune a related source model to a target dataset. For example, Maqsood et al. (2019) proposed a transfer learning method to predict Alzheimer's disease by finetuning a modified AlexNet (Krizhevsky et al., 2017; Zhuang et al., 2020). Shin et al. (2016) applied a finetuning approach to transfer knowledge from non-medical to medical image data domain for computer-aided detection tasks (Weiss et al., 2016). We include a finetuning approach as a baseline and find that our proposed approach outperforms it.

Advanced transfer learning methods have also been used in this field, for example, Yoon et al. (2022) proposed a zero-shot transfer learning method for Heterogeneous Graph Neural Networks that transferred learning from label-abundant node types to label-scarce node types for a classification task. Their approach would not be applicable to our case as their approach can be used only in the case where the node type shares the same task and in our case, we are transferring knowledge between different tasks. Geyer et al. (2019) proposed a transfer-learned model with multiple source models for medical image segmentation. They provide a mechanism using transfer learning and lifelong learning for conceptually combining source models that still perform well on all prior tasks. In our approach, we are not focused on how well our model performs on the prior tasks, moreover, we are not using lifelong learning rather we are using QD optimization technique to improve on the already finetuned model. There are other domain-specific approaches of transfer learning in the medical domain which we do not cover here due to their lack of generality.

## 4 OUR APPROACH

In this section, we present Quality-Diversity Transfer Learning (QDTL) method. Our approach is based on domain transfer where we have distinct source and target domains and the goal is to transfer knowledge from the source to the target domain. QDTL can be defined as a three-step process. First, we train a neural model on the source domain data; this step is domain-specific. Then the weights of the trained model are used to finetune it on target domain data. Finally, we take that model and run quality-diversity over the model weights to find the most optimal model for our problem. To the best of our knowledge, this is the first time QD has been applied as neuroevolution for supervised transfer learning.

### 4.1 ARCHITECTURE

We use a bidirectional Long Short Term Memory(Bi-LSTM) Recurrent Neural Network-based architecture as our source model, consisting of 4 Bi-LSTM layers. Each of the 4 Bi-LSTM layers contains 512 units with a dropout size of 0.2. They use the default linear activation function. These 4 Bi-LSTM layers are followed by a dense prediction layer using the default linear activation function at the end with a unit size of 1. We employed Keras and used all the default values for its Bi-LSTM and Dense layers otherwise. A small model is used because of the lack of training data and to avoid overfitting. Further, RMSprop is used as optimizer and mean square error (MSE) for calculating loss for both step one where we train our model on the source data and step two, where we finetune the model on our target data. We call the model from step one as the source model and from step two as the target model. Since we are predicting numerical values, we chose MSE as our loss function. For our target model which is the basis for our approach and all baselines and ablations, we use all the same parameters as our source model but instead of 4 Bi-LSTM layers, we use 4 LSTM layers. We used the bidirectional LSTM model for our source model because it has a larger dataset size, moreover, the data is not sequential. The Bi-LSTM model works better with larger datasets and helps us figure out larger-range dependencies in the case of the source model. For our target model, we used an LSTM model as it allows us to compare it to CE-MCTS more easily (Mahajan & Guzdial, 2022).

### 4.2 QUALITY-DIVERSITY TRANSFER LEARNING

Quality-Diversity (QD) is an evolutionary algorithm, which involves generating and evolving a population of models over many generations. QDTL uses QD to optimize the performance of a DNN model trained on a source task for our target task. QD is used over other optimization algorithm to achieve a variety of models to reduce the risk of overfitting. We hypothesized this would allow for higher performance of our final DNN models for our low-data tasks (Floreano et al., 2008). Further, each generated model is evaluated based on quality and diversity fitness objectives.Our approach assumes two objective functions $f_Q : R^n \to R$ and $f_D : R^n \to R$ for quality and for the diversity populations respectively. For each model $s_Q \epsilon S_Q$ and $s_D \epsilon S_D$, where $S_Q$ is our generated quality population and $S_D$ is our diversity population, the goal of our approach is to select models from these population based on their respective objective functions and then select the top 10 models based the quality fitness function. We finally return the single best model.

QDTL method is shown in Algorithm1. First we train a model on the source dataset using Cervical Cancer dataset[1] with 858 samples. We chose this particular dataset as our source dataset due to the domain similarity and some feature similarity with our target domains. For example, demographic information and historic medical data are similar features between our source and target datasets. This dataset has 46 input features. We employed a 60/40 train-validation split, we used this split to make the size of the training set closer to the target dataset. We train the neural network architecture described above for 100 epochs, with a learning rate of 0.0001. We determined all hyperparameter values based on the validation set. We further finetune this model on training data from our target task. This is our baseline model called "finetune". This target model is given as input to a quality-diversity process, in which we optimize the weights directly according to the target training dataset. Line 1 represents this input model.

---

[1] https://www.kaggle.com/datasets/loveall/cervical-cancer-risk-classification

We run our QD architecture search and transfer process to output the final target model. Based on the input, we initialize two populations (i.e., quality population and diversity population) of fixed size by running our mutation function $popsize - 1$ times. As with other evolutionary algorithms, QD requires specialized mutation, crossover, and fitness function (details in subsections below). Our algorithm combines both crossover and mutation operations as neighbour functions to optimize the neural network architecture and parameters. The crossover operation involves combining layers from two parent networks, and the mutation operation introduces random changes to the parameters of a network. After performing the crossover process by combining layers from the parents, there's a chance that mutation will occur based on the predefined mutation rate (mutationRate). In this work, we used a mutationRate of 0.2 meaning there was a 20% chance that a mutation function was called to introduce small random changes to the architecture of the child network. We run for 20 generations, then select the top 10 models according to our quality fitness and then return the final best model. We chose 20 generations and other hyperparameters based on the validation performance of our first problem domain (Section 5.1).

---

**Algorithm 1** QDTL Approach

**Input:** An architecture $A$, the population size $pop\_size$, maximal generations $gen$, the $source$ dataset, and the $target$ dataset.
**Output:** Best performing architecture.
1  $A \leftarrow$ train $A$ on $source$
2  $pop_Q = \{A\}$
3  $pop_D = \{A\}$
4  **while** $|pop_Q| < pop\_size$ **do**
5     $network_Q \leftarrow$ Mutation($A$)
6     $network_D \leftarrow$ Mutation($A$)
7     $pop_Q$.append($network_Q$)
8     $pop_D$.append($network_D$)
9  **end**
10 i $\leftarrow 0$
11 **while** $i < gen$ **do**
12    $pop \leftarrow$ Crossover($pop_Q$, $pop_D$)
13    $pop \leftarrow$ Mutation($pop_Q$, $pop_D$, $mutationRate$)
14    $pop \leftarrow$ Reduce($pop$, $fitness\_Q$, $fitness\_D$)
15    i $\leftarrow$ i + 1
16 **end**
17 architecture = Max($pop_Q$)
18 return architecture

---

### 4.3 CROSSOVER

In quality-diversity search, the crossover function combines genetic information from two or more parent models to create a new offspring model that explores different regions of the search space and promotes diversity among the models. We conduct crossover between both populations. First, we select two parents via a weighted sampling based on the quality and diversity of fitness scores. Second, we identify the position of the LSTM layers in both selected parents. Then we randomly choose one position in the LSTM layers of both parents. We only track LSTM layers as these are the layers in which feature extraction occurs. Using this position, we take the initial half of the weights and layers from the first model, and the latter half of the weights and layers from the second model. By creating new models from the quality and the diversity population, we hope to combine the good qualities of existing models and promote diversity in the search space.

### 4.4 MUTATION

We have eleven mutation functions and randomly select one of them for our mutation process. Conceptual expansion (CE) is utilized in our mutation functions, which involves broadening the scope of a concept beyond its original meaning, in our case that is the model weights. By representing a specific weight as a combination of $\alpha$ and $f$, we are able to discover high-quality weight combinations. Directly modifying the $\alpha$ and $f$ values associated with each weight allows us to manipulate the network weights as represented using the equation

$$CE(F, \alpha) = \alpha_1 * f_1 + \alpha_2 * f_2 + ... + \alpha_n * f_n \tag{1}$$

Our first four functions are the ones used for conceptual expansion in the work by Guzdial & Riedl (2019). These were successfully employed by Mahajan & Guzdial (2022) for the transfer learning approach. We use the same four functions in our approach. The first function multiplies a random element in a chosen $\alpha$ matrix by a scalar from [-2, 2]. The second scales all elements in a selected $\alpha$ matrix similarly. The third randomly swaps pairs of $\alpha$ and $f$ values with matching dimensions, introducing variability. Lastly, the fourth adds two randomly chosen $\alpha$ and $f$ values (with equivalent dimensions) to a CE approximation. For our remaining seven functions, we take inspiration from the work by Singamsetti et al. (2021) and in addition to using two of their mutation functions which are, adding random $\alpha$ and $f$ values to a random position to a random mutation layer, we extend these by using subtraction, multiplication and division to a random layer and index.

### 4.5 FITNESS SCORE

We use two fitness scores, one for the quality population and the other for the diversity population. For the quality fitness function, we evaluate the mean square error on the training set. For the diversity population, we use exploratory fitness. For this, we create a matrix of the weights of the input model which is represented $W_B$ and take the absolute difference with the matrix of weights of the child model which is represented by $W_C$. We return the mean of this difference. We use this exploratory fitness function to find models in the search space that are far apart from our input model.

## 5 PROBLEM DOMAINS

We evaluate QDTL on two tasks in the medical domain. For the first task, we predict the gestational age of pregnant women in order to predict preterm birth (PTB). Gestational age is the number of days after which the woman will deliver. For the second task, we conduct a survival analysis of patients who have undergone organ transplants and we predict the number of days after the transplant that these patients survive (up to a final check-up). Since both delivery date prediction and survival days after an organ transplant are more complex functions that cannot be reduced to a simple linear or logistic function, we believe DNNs are better suited to solve this problem. To demonstrate this, we performed logistic regression on one of our tasks and compared its performance with that of finetuning. It surpassed logistic regression. Due to space limitations, we have included these results in the supplementary material. One of the most basic requirements of modern DNNs is a large dataset. However, there are limited samples in both cases, and a naive application of DNNs would not provide an effective solution. We, therefore, hypothesized that a transfer learning approach would be better suited to solve these problems. It is important to note that we obtained informed consent to utilize this data for our research purposes.

### 5.1 PRETERM BIRTH (PTB)

 PTB is defined as birth before 37 full weeks of gestation. Preterm birth is the leading health problem during the perinatal period, the leading cause of death in children aged under 5, and a major cause of chronic disease (Romero et al., 2014; Chang et al., 2013; Patel, 2016). A diagnostic approach for predicting preterm delivery is the need of the hour as it will help communities worldwide and will stimulate the creation of treatments against preterm birth. Many clinical laboratory tests are also expensive and therefore not attainable for those populations most at risk for PTB (Heng et al., 2016; Moufarrej et al., 2022; Scott et al., 2020). Moreover, the existing DNN approaches would require a lot more data than what is available for PTB due to data availability and privacy concerns. Our approach not only tries to overcome the low-data limitation problem but predicts women at risk for a PTB, and actually predicts when these women will deliver. This output would be helpful in designing appropriate treatment for these women.

For this problem, we had 70 features and 135 total samples. We got these 135 samples from Christiaens et al. (2015). Where 43 samples were of women who had PTB and the remaining 92 samples of women who had normal delivery or non-preterm birth (NPTB) samples. Since we had two sets of data PTB and NPTB and there is a data imbalance between the two datasets. Using a

combination of both and finetuning the source model trained on the cervical cancer dataset twice helps to reduce the data imbalance. First, we finetune the model on an NPTB dataset and then further finetune on a combination of PTB and NPTB samples for 30 epochs each with a learning rate of 0.0001. This model then serves as the input to our QD approach.

## 5.2 ORGAN TRANSPLANT

Solid organ transplantation allows people with terminal organ failure and no other treatment options to receive a donor organ from another human. In our case, samples of children requiring heart, kidney or liver transplants have been collected in a national collaboration before and 3 and 12 months after transplantation. We predict days to the endpoint, which is the number of days after an organ transplant the patient survives censored at the end of the observation period. For the organ transplant problem, we had 130 samples in this task and 158 features. For this task, since we have a single dataset, we only finetune the source model which is trained on the cervical cancer dataset once for 45 epochs, with a learning rate of 0.001.

# 6 EXPERIMENTS

In this paper, we try to overcome the low-data limitation problem in the medical domain by utilizing transfer learning and quality-diversity optimization. We evaluate QDTL on two tasks from the medical domain using five-fold cross-validation.

For Preterm Birth (PTB) problem set, we predict the gestational age. Since we had two sets of data on which we were finetuning our base model, we created minifolds in addition to five-fold cross-validation to test out all combinations of the two datasets. For example, for fold1 we use 80% of NPTB data for training the model. We represent this 80% data as A, B, C and D where each represents 20% NPTB data. We interchangeably choose 3 folds out of these and use them for the initial finetuning of the source model. The remaining one fold is combined with 80% PTB data and used for the final finetuning of the model we get from the previous step. We repeat this step 4 times, each time combining one of the NPTB folds with 80% PTB data. We repeat this for each of the remaining 4 folds so as to make sure our model is robust. For our second organ transplant task we did a simple 5-fold cross-validation, with an 80/20 train-test split.

We used fixed seeds for all tests and computed the average mean square error and standard deviation on the test dataset to evaluate the performance of different approaches. All of the baseline approaches, ablations and our QDTL approach are executed roughly for around 4 hours independently. These experiments are conducted using 6 CPUs and 2 NVIDIA Tesla V100 GPUs per task.

## 6.1 BASELINES AND ABLATIONS

In this work, we employed two baselines and created four ablations. For all the ablations we use the same eleven mutation functions as QDTL and we run each for the same 20 iterations for a fair comparison. For the organ transplant task, we only use the two baselines for comparison as the PTB task already shows the relative performance of the ablations. For the baselines, it was essential to compare QDTL with a finetuning baseline given its prevalence. We additionally show the performance on a no-transfer learning baseline as this is the more common approach.

### 6.1.1 BASELINES

- The first baseline approach, which we refer to as "No-Transfer Learning," involves directly training the model on the training dataset without any transfer learning. For the PTB task, we trained the model for 30 epochs, while for the Organ Transplant task, we trained it for 45 epochs. We used learning rates of 0.0001 and 0.001, respectively. We selected these hyperparameters based on preliminary experiments conducted to optimize the training process.

- The second baseline, called "Finetune", involves finetuning our source model on the target domain data. This represents the standard approach one might take to solve this type of transfer learning problem for a medical prediction task (Maqsood et al., 2019; Shin et al.,

2016). We utilized this model as an input to all other baselines, ablations, and our QDTL approach, making it an additional ablation of our approach.

### 6.1.2 Ablations

- The first ablation is called "CE-MCTS" and is adapted directly from Mahajan & Guzdial (2022). The difference here is in the use of MCTS over QD. We used their same setup but executed their approach with 20 iterations of 10 rollouts of length 10.
- The second ablation, referred to as "Random Walk," employs a random walk approach to explore the model space instead of QD. In this technique, at each step, a random child is selected, and this process is repeated for 20 iterations. The selection of the best model is based on the quality fitness function criteria used in QDTL, facilitating a straightforward comparison between the techniques.
- The third ablation, referred to as "Greedy," employs a greedy search strategy instead of QD. In this technique, the mutation function described earlier is utilized to generate ten random neighbours. The neighbour with the highest node value is selected at each step, and this process is repeated for 20 steps. The best final model is chosen based on the quality fitness function criteria used in our QDTL approach.
- The fourth ablation, known as "GA," employs a genetic algorithm with the same configuration as our QDTL approach but without the diversity population and diversity fitness score. The purpose of including this ablation was to demonstrate that the dual optimization in QD allows for a more beneficial exploration of the search space.

## 7 Results

We present the results in the form of the average number of days off from the actual for two tasks: preterm birth (PTB) and organ transplant. The average number of days is computed across different folds in each task. Additionally, since PTB problem domain consists of minifolds, we also provide the standard deviation (SD) values across all baselines for each of the minifolds within this task.

To calculate the average number of days for the PTB task, we computed the actual values for each minifold within a fold. For example, to calculate the average number of days for fold1, we computed the actual values for each of the four minifolds: fold1A, fold1B, fold1C, and fold1D. We then took the average of these four values and compared them to the actual test values, computing the absolute difference. We then divided this absolute difference with nine, the number of samples in this particular fold. The test dataset contains 9 samples for fold1, fold2, and fold3, while for fold4 and fold5, there are 8 samples each.

For the organ transplant task, since there are no mini-folds, we directly computed the absolute difference between the actual and predicted values and divided this difference with 26, which is the number of samples in each fold. Table 1 and Table 2 show results for the PTB task and organ transplant task respectively.

For the PTB task results, QDTL overall performs better than the other baseline and ablation approaches. In particular there is an approximate difference of 2 days between QDTL and other approaches for fold2, and fold5, and of 1 day in case of other folds. Upon analysis we found that this reduced difference of 1 day between QDTL and other approaches is attributed to the missing data in case of fold 1, 3, and 4. This shows QDTL, in general, performs well at the task of transfer learning.

Similarly, in case of the organ transplant task, QDTL performs better than the other other two baselines. Noticeably, in fold5 QDTL outperforms No-transfer learning by a margin of 19 days and outperforms Finetuning by 28 days. Our approach appears to be the most effective and consistent in predicting the Organ Transplant Task, as it consistently yields the lowest average prediction errors and has the lowest standard deviation among the three approaches. It is important to note that the numbers reported for this task are relatively high due to the nature of the data, where we are predicting over years instead of within a single year.

In this task, each test fold consists of 26 samples, which were used for evaluating the performance of the different approaches. The QDTL approach demonstrates superior performance compared to the other baselines, indicating its effectiveness in predicting the survival days for organ transplant

Table 1: Average number of days by which the model is off over five cross-validation folds of PTB Task for QDTL and other approaches.

| Approach | Fold1 | Fold2 | Fold3 | Fold4 | Fold5 | Average |
|---|---|---|---|---|---|---|
| No-Transfer Learning | 9.18±0.65 | 3.06±0.064 | 17.28±0.83 | 4.91±0.26 | 18.8±1.17 | 10.65 |
| Finetune | 9.27±0.62 | 3.05±0.06 | **16.85±1.11** | **4.9±0.17** | 21.14±1.76 | 11.05 |
| CE-MCTS | 9.2±0.65 | 3.13±0.03 | 17.19±1.33 | 5.27±0.32 | 21.95±3.04 | 11.35 |
| Random Walk | 9.24±0.61 | 3.13±0.18 | 17.24±1.19 | 5.26±0.16 | 20.49±1.42 | 11.07 |
| Greedy | **9.18±0.69** | 3.06±0.18 | 17.29±1.13 | 5.3±0.16 | 19.99±0.9 | 10.96 |
| GA | 9.25±0.64 | 3.16±0.06 | 17.23±1.46 | 5.22±0.17 | 21.09±1.52 | 11.19 |
| QDTL | 9.38±0.39 | **2.79±0.76** | 17.89±0.26 | 5.18±0.23 | **18.38±0.21** | 10.72 |

Table 2: Average number of days by which the model is off over five cross-validation folds of Organ Transplant Task for QDTL and other baselines.

| Approach | Fold1 | Fold2 | Fold3 | Fold4 | Fold5 | Average |
|---|---|---|---|---|---|---|
| No-transfer Learning | 169.660 | **213.301** | 216.500 | 168.357 | 266.028 | 206.769 |
| Finetune | 167.057 | 214.798 | 226.761 | 185.142 | 275.055 | 213.763 |
| QDTL | **162.697** | 213.485 | **216.080** | **163.765** | **246.677** | 200.541 |

patients across most folds. It is observed that in fold2, No-transfer learning approach predicts better than QDTL by approximately one day. However, the difference is much larger between QDTL and other approaches in the cases where QDTL outperofrms them.

## 8   LIMITATIONS AND FUTURE WORK

This paper presents a novel approach called Quality-Diversity Transfer Learning (QDTL) to address the challenge of limited data in medical domain prediction tasks. The results demonstrate the efficacy of our approach in this domain, suggesting the potential for it to address low-data problems in other domains. However, further exploration is needed. In addition, the performance of our approach is currently dependent on the finetuned model used as input, and the finetuning process requires careful consideration of the dataset, model architecture, and other features of the source and target domains. This dependence on the finetuning process creates challenges in terms of choosing various hyperparameters, which we plan to investigate in future work. Overall, our study contributes to the ongoing research on addressing limited data challenges in medical prediction models and paves the way for exploring the application of QDTL in other low-data domains.

## 9   CONCLUSIONS

In this work, we present a novel approach called Quality-Diversity-Transfer Learning (QDTL) to solve low-data problems. This approach relies on a combination of transfer learning, architecture search and an evolutionary approach. We evaluate the effectiveness of our approach by comparing it with standard baselines and ablations on two medical prediction tasks. Our results indicate that QDTL outperforms the baselines and ablations in terms of efficiency, resulting in higher-quality models more closely able to predict desired output features. This demonstrates the potential of our approach to improve model performance on various low-data tasks.

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
