# OpenReview forum: "Quality-Diversity Transfer Learning (QDTL)"
_ICLR.cc/2024/Conference — ICLR 2024 Conference Withdrawn Submission_

### Official Review · Reviewer_Mn8Q · 2023-11-03

**Soundness:** 2 fair
**Presentation:** 2 fair
**Contribution:** 2 fair
**Rating:** 3
**Confidence:** 4

**Summary:**

The paper proposes a neuroevolutional approach to deal with small datasets.

**Strengths:**

- There is certainly a scope in improving transfer learning for small datasets, and one of the possible ways to do it is to use neuroevolutionary approaches

**Weaknesses:**

However, there are a number of questions which need to be answered to make it a strong contribution.

- on originality and significance: there have been plethora of papers [1,2,3] using genetic algorithms for optimising neural networks in one form or another. As things stand, it is not clear what is exactly methodological contribution in contrast with these approaches. In particular, [3] proposes the combination of genetic algorithm with gradient-based local search which seems to be related. One could think of the proposed algorithm 1 as a combination of existing methods, and it is important to show why this combination is not trivial.
- on clarity: there are a number of questions both on the paper structure and reproducibility (see Q1, Q2, Q3)
- on quality: the chosen motivation of medical prediction may sound unconvincing (see Q4) because the proposed medical problems would require consideration of causal relationships to make practical sense (see, for example, the difference with paper [4] which estimates (causal) treatment effects.

[1] Stanley, Kenneth O., et al. "Designing neural networks through neuroevolution." Nature Machine Intelligence 1.1 (2019): 24-35.

[2] Montana, David J., and Lawrence Davis. "Training feedforward neural networks using genetic algorithms." IJCAI. Vol. 89. 1989.

[3] D’Angelo, Gianni, and Francesco Palmieri. "GGA: A modified genetic algorithm with gradient-based local search for solving constrained optimization problems." Information Sciences 547 (2021): 136-162.

[4] Bica, Ioana, and Mihaela van der Schaar. "Transfer learning on heterogeneous feature spaces for treatment effects estimation." Advances in Neural Information Processing Systems 35 (2022): 37184-37198.

**Questions:**

Q1 The paper's structure and presentation could be improved. For example, the first sentence of Section 6 is: "In this paper, we try to overcome the low-data limitation problem in the medical domain by utilizing transfer learning and quality-diversity optimization. " Isn't it something that we already know by this time? Then it can be safely removed.  Not so many works, despite the topics of the paper being largely popular, are covered in the Related works section.  The experimental conditions and hyper parameters are split between sections 4.2 and 5.1 making it difficult to understand.

Q2 Reproducibility of the paper can be improved. For example, I couldn't find where and how the authors select popsize parameter and whether the same exact parameters have been used throughout all experiments.

Q3 In Table 1, it is unclear whether there is a statistically significant improvement from using the proposed methods. It would be great if the authors could clarify upon this.

Q4 The paper presents the findings on a number of problems which belong to the medical domain and usually considered as a problem of treatment effects estimation. In many standard machine learning questions (e.g., classification of ImageNet data), fitting classifier or regression model to get a functional dependency model is enough; however, I don't think this is enough for the proposed medical problems. In these problems, we need to distinguish between spurious correlations and causes and consequences between inputs (see Figure 1) and the predictions. Bica et al (2022), for example, make this important distinction and shows how the method could be used for predicting treatment effect.

---

### Official Review · Reviewer_vX3E · 2023-11-04

**Soundness:** 2 fair
**Presentation:** 2 fair
**Contribution:** 2 fair
**Rating:** 5
**Confidence:** 3

**Summary:**

The paper introduces a solution called Quality-Diversity Transfer Learning (QDTL) to tackle the
challenging issue of limited data in medical prediction tasks. It applies QDTL to two medical scenarios: predicting
preterm birth and the survival days of organ transplant patients. The experiments comparing QDTL against
standard methods demonstrate QDTL's superior performance. This proposed approach
combines transfer learning, architecture search, and evolutionary methods, incorporating quality-diversity
optimization and mutation functions to make it stand out.

**Strengths:**

The paper effectively conveys the significance of the problem – dealing with limited data in medical predictions, a critical issue in healthcare. The paper effectively demonstrates QDTL's superiority by consistently outperforming baseline methods, showcasing its potential in handling low-data medical prediction tasks.

**Weaknesses:**

Though the overall solution looks reasonable, most of the components seem to be existing general ideas and not original from this paper. Therefore, it's not clear what are the essential technical contribution. Also, I feel the main idea of this paper lies in neural architecture search rather than transfer learning (emphsized in the title). But the adopted NAS algorithm is not particularly designed for the fine-tuning scenario.

**Questions:**

1. The paper could provide more insights into addressing missing data, which might impact results.
2. Offering specific guidance on hyperparameter optimization during the finetuning process could enhance
practicality.

---

### Official Review · Reviewer_FuMo · 2023-11-06

**Soundness:** 2 fair
**Presentation:** 2 fair
**Contribution:** 1 poor
**Rating:** 3
**Confidence:** 3

**Summary:**

The paper presents a new approach to transfer learning, termed Quality-Diversity Transfer Learning (QDTL), which employs neuroevolution techniques to address the challenge of transfer learning to target domains with low-data.

**Strengths:**

- The paper presents a interesting new approach to tackle low-data problems in transfer learning.

**Weaknesses:**

- Motivation: The approach (application of QD methods to transfer learning) is not very well motivated.
And how is it related to dealing with low-data regime in particular?
The method seems like a random mix of ideas in ML without much rationale.

- Generalizability: The evaluation of method's performance is limited to two medical prediction problems. While the results are promising, more extensive evaluation on standard settings (e.g., ImageNet transfer) would be valuable.

- Complexity and Scalability: The approach seems computationally intensive. The authors should address the computational costs associated with QDTL, including training time and resource requirements, and compare to that of baselines.

**Questions:**

- Why is neuroevolution a reasonable approach for this problem?

---

### Official Review · Reviewer_JVCK · 2023-11-09

**Soundness:** 3 good
**Presentation:** 3 good
**Contribution:** 3 good
**Rating:** 3
**Confidence:** 2

**Summary:**

This paper introduces Quality-Diversity Transfer Learning to deal with low-data problems in medical tasks. It consists of source domain training, target domain finetuning and quality-diversity model selection. The authors conduct the experiments on two medical problems to verify the effective of their approach.

**Strengths:**

Low data is common in the medical domain. Improving the transfer learning is one of the promising directions.

**Weaknesses:**

The improvement is not significant in the Organ Transplant Task. The proposed method is more complex and time-consuming but is comparable with the no-transfer learning baseline. This approach is not that robust to very small datasets.

The novelty of this work is limited. The pipeline described in Fig. 1 is very simple.

**Questions:**

please see my comments on the weaknesses.